# The Efficacy of Tele-Rehabilitation Program for Improving Upper Limb Function among Adults Following Elbow Fractures: A Pilot Study

**Naomi Mayer [1,2], Sigal Portnoy [1,*] , Ram Palti [3] and Yafa Levanon [1,2]**

[1] Department of Occupational Therapy, Sackler Faculty of Medicine, Tel Aviv University, Tel Aviv 6997801, Israel; naomi.z29@gmail.com (N.M.); yafil@tauex.tau.ac.il (Y.L.)
[2] Occupational Therapy Department, Sheba Medical Center, Ramat Gan 52621, Israel
[3] Hand Surgery Department, Sheba Medical Center, Ramat Gan 52621, Israel; ram.palti@gmail.com
[*] Correspondence: portnoys@tauex.tau.ac.il; Tel.: +972-3-6405441; Fax: +972-3-6409933

**Abstract: Background**: Active mobilization post-elbow fractures reduces the incidence of complications. Occupational therapists use tele-rehabilitation, incorporating technology into their practices. There is a lack of evidence-based trials regarding the integration of tele-rehabilitation during treatment. We therefore aimed to compare tele-rehabilitation treatment outcomes with conventional rehabilitation in improving the upper limb function post-elbow fractures. **Methods**: Eighteen participants post-elbow surgery due to fracture were divided into two groups according to age and fracture type. The groups received one month of treatment: the tele-rehabilitation group (N = 9, median age 33.0 ± 27.9 years, range 18.5–61.0) received 1–2 tele-rehabilitation treatments per week via a biofeedback system of elbow motion (the ArmTutor and 3D Tutor systems, MediTouch Ltd., Netanya, Israel) and 1–2 treatments in an outpatient clinic, and the control group (N = 9, median age 60.0 ± 37.0 years, range 20.5–73.0) received 3–4 treatments per week in the clinic. Both groups were instructed to self-practice at home. Four evaluations were performed: before and after the intervention, and 3 months and 1 year from surgery. The outcome measures included the Jebsen–Taylor hand function test; the disabilities of the arm, shoulder, and hand questionnaire; the patient-rated elbow evaluation; satisfaction; passive and active range of motion (ROM); and strength measurements. **Results**: Findings demonstrated a significant improvement in the ROM and in functional assessments in both groups. No statistically significant differences were found between the groups. The subjects in the tele-rehabilitation group reported a higher level of satisfaction and needed less help from a family member during practice. **Conclusions**: Tele-rehabilitation programs could be incorporated in the framework of treatment following elbow fractures. Tele-rehabilitation is a cost-effective treatment, suitable for patients with accessibility difficulties or who have difficulty arriving at the clinic.

**Keywords:** biofeedback; arthrolysis; tele-rehabilitation; elbow

## 1. Background

The elbow is an anatomically complex joint prone to complications and functional impairments following trauma [1]. Complications may include limitation in range of motion (ROM), nerve injury, pain, joint instability, formation of heterotopic ossification, and joint stiffness [2]. Limitation to the functionality of the elbow may affect everyday activities [3,4]. It is estimated that a 50% decrease in elbow function may lead to an 80% reduction of daily activities [5]. Rehabilitation following elbow injury is complex and poses challenges for both the patient and the therapist [6]. It was shown that prolonged immobility following elbow injury is a major factor in limiting joint ROM [7]. Most evidence-based literature supports early joint activation in order to restore limb function [1,8] In the topic of elbow rehabilitation after fractures and dislocations, there are few reports of the efficacy of active interventions, and most of the protocols are based on clinical experience [1,4,9,10].

Tele-rehabilitation is a tool that enables clinicians to perform evaluation and treatment of the patient using communication technologies. Studies show that tele-rehabilitation is effective for recovery of motor function following stroke or traumatic brain injuries [11–13]. Unfortunately, studies investigating the efficacy of tele-rehabilitation following orthopedic injuries are scarce [14,15]. Moreover, within the literature concerning tele-rehabilitation of orthopedic injuries, most concern lower limb injuries. A recent review surveying gaming and tele-rehabilitation for treating orthopedic injuries [16] showed that the great majority of tele-rehabilitation studies included lower limb orthopedics patients. While advanced technologies for allowing monitored elbow tele-rehabilitation emerge [17,18], literature investigating the value of tele-rehabilitation following elbow injuries is lacking. To the best of our knowledge, while tele-rehabilitation was shown to be a valid and reliable tool for assessment of elbow disorders [19], no previous studies investigated the effect of tele-rehabilitation following elbow fractures. It was suggested that some of the advantages of tele-rehabilitation are that it reduces health care costs, improves treatment adherence, and is delivered in a manner that is satisfactory to the patients [20–22]. However, evidence-based trials quantifying the integration of computerized devices in tele-rehabilitation for motor rehabilitation are lacking. The COVID-19 pandemic that resulted in government regulation of social distancing has increased the need for solutions for remote healthcare [23,24].

Considering all of the above, the aim of this study was to compare quantitative objective and subjective measures of rehabilitation following elbow fracture, between a group who received conventional treatment and a group who received a combination of tele-rehabilitation via a biofeedback device and conventional treatment. The two groups received one month of intervention immediately post-surgery and were evaluated at four time points: immediately after the surgery, 1 month post-surgery, 3 months post-surgery, and one year post-surgery. Both follow-up trials performed, 3 months and 1 year post-surgery, were conducted in order to detect postintervention effects, conveying information about the long-term benefits recorded in each group.

## 2. Methods

### 2.1. Participants

The study design is presented in Figure 1. Eighteen adults participated in this study. Inclusion criteria: age range of 18–80, post-Open Reduction and Internal Fixation (ORIF) of the elbow or arthrolysis surgery, stable for mobilization. Exclusion criteria: neurological or orthopedic disease that could affect upper extremity function. This study was approved by the hospital's Helsinki committee (approval number SMC-2296–15) and the university's ethics committee. The subjects were assigned to one of two groups (Table 1) and matched according to age and type of fracture (olecranon, distal humerus, radial head, or arthrolysis): tele-rehabilitation group (N = 9) and control group (N = 9).

**Table 1.** Personal characteristics of the participants in the tele-rehabilitation and control groups. Values are presented as median and interquartile range in parentheses.

| | Tele-Rehabilitation Group (N = 9) | Control Group (N = 9) | *p* |
|---|---|---|---|
| Age (years) | 33.0 (27.9) | 60.0 (37.0) | 0.185 ¥ |
| Sex | 3 male; 6 female | 4 male; 5 female | 1.000 § |
| Injured hand | 3 Right; 6 Left | 3 Right; 6 Left | 1.000 § |
| Injured hand | 3 D; 6 ND | 3 D; 6 ND | 1.000 § |
| Hand dominance | 7 Right; 2 Left | 8 Right; 1 Left | 1.000 § |
| Fracture type | 4 Ol, 3 DH, 2 Arth | 4 Ol, 3 DH, 2RH, 1 Arth. | 0.469 § |

¥ Mann–Whitney test, § Chi-Square test; D = dominant; ND = nondominant; Ol = Olecranon; DH = Distal humerus; RH = Radial head; Arth. = Arthrolysis.

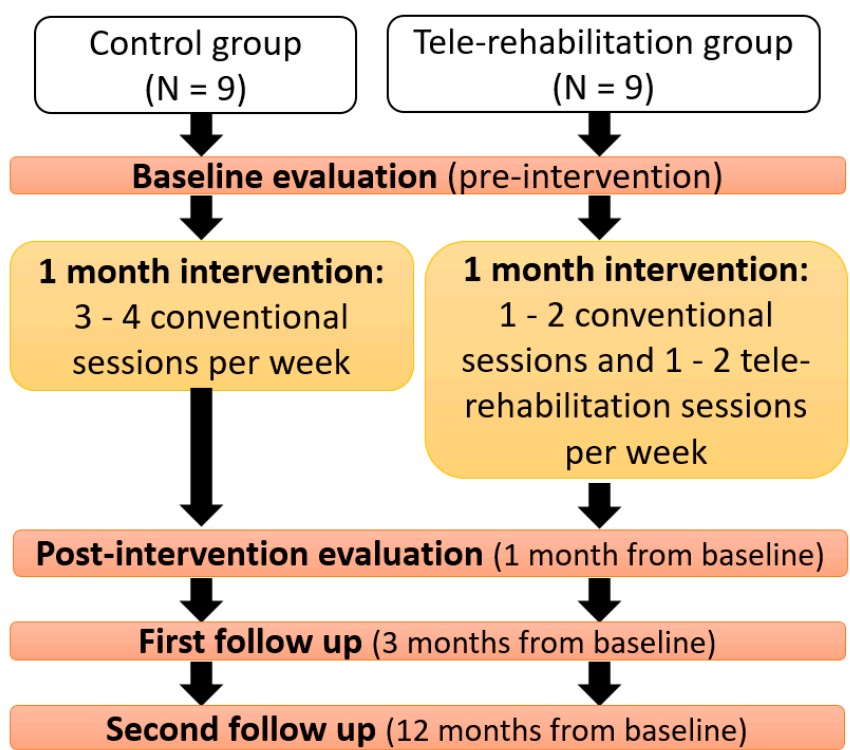

**Figure 1.** The outline of the study.

*2.2. Study Tools and Protocol*

2.2.1. Evaluation

We used a demographic questionnaire and retrieved data from medical files to obtain age, gender, occupational performance before injury, and other medical conditions. Then, four evaluations were performed: immediately after the surgery, after one month of intervention, three months post-surgery, and one year post-surgery.

Three physical outcome measures were acquired in each evaluation: *Active and passive ROM* measured using a manual goniometer and *grip strength* measured in kilograms using a hydraulic hand dynamometer (Jamar, Duluth, MI, USA). The subjects performed three trials while sitting in the same position (elbow in 90° flexion and arm in mid-position) following standard instructions [25,26]. The mean of the three measurements was calculated; *The Jebsen–Taylor Hand Function Test* (JHFT; [27]) which is a standard performance assessment of hand function in seven day-to-day activities, e.g., lifting small objects and eating. Each task is measured by time and performed first by the nondominant hand and then by the dominant hand.

Four questionnaires were filled out by the subjects in each evaluation:

1. *The Disabilities of Arm, Shoulder and Hand* (DASH; [28]), a self-assessment functional questionnaire used following an upper extremity injury. The questionnaire includes 30 items concerning functioning and severity of the symptoms in the past week. The patient is asked to rate the difficulty level of performing 21 physical activities on a Likert scale (from 1 = no difficulty to 5 = unable to perform) and the influence of the physical injury on the social functioning (1 = not at all to 5 = extreme). The final score ranges from 0 to 100, with a higher score indicating poorer functioning and more severe symptoms. DASH has a high repeatability in subjects with elbow injuries (ICC = 0.92; [29]). It has also been found that the tool is sensitive to post-treatment status changes compared to the pretreatment condition [29].

2. *The Patient-Rated Elbow Evaluation* (PREE; [30]), a self-assessment questionnaire following elbow injuries which evaluates the level of pain and level of function of the affected limb on a Likert scale between 0–10 (0 = no pain and no functional limi-

tations to 10 = the highest pain or highest disability). The questionnaire has three parts: 5 questions are related to the level of pain; 11 questions regarding the ability to perform specific tasks with the elbow, e.g., combing the hair; and 4 questions related to usual daily life activities (dressing, eating, etc.). High scores reflect higher pain and disability. This questionnaire was translated from English to Hebrew as part of this study in a re-translation procedure and expert validation.

3. *Patient satisfaction questionnaire*, *about self-management of the home program*, a self-completion questionnaire compiled by Dr. Debbie Rand from the Department of Occupational Therapy at Tel Aviv University and adapted for this study. The questionnaire consists of 7 questions. Four items used a Likert scale between 1 and 5 (1 = not at all and 5 = very much): satisfaction from the intervention, level of motivation evoked by the intervention, subjective motor improvement of the injured hand, and the duration of the practice. The other 3 items regarded the assistance needed to use the intervention at home (needs help all the time, occasionally needs help, or do not need help at all) [31].

4. *The System Usability Scale* (SUS; [32]), a questionnaire filled out by the subjects in the tele-rehabilitation group, intended to examine the degree of convenience of using technological systems. This questionnaire consists of 10 statements on the Likert scale (1 = strongly disagrees to 5 = strongly agrees). The scoring is performed as follows: uneven items (1, 3, 5, 7, 9) are deducted by one point from the subject selection, and even items (2, 4, 6, 8, 10) subtract the subject choice from five. The sum of the items is multiplied by 2.5, and a score is obtained on a scale between 0 and 100. A higher score indicates a more positive opinion about the comfort of the system.

### 2.2.2. Intervention

Each group received one month of intervention, which commenced immediately following the surgery. The tele-rehabilitation group received 1–2 sessions per week of elbow motion at home via tele-rehabilitation and 1–2 conventional sessions per week in an outpatient clinic. The control group received 3–4 conventional sessions per week in the outpatient clinic. Each session lasted 30 min. Additionally, both groups were instructed to perform home exercises. The tele-rehabilitation group was instructed to use the MediTouch system, and the control group was instructed to repeat the exercises that they performed in the clinic. After one month, both groups continued to receive occupational therapy treatment in the clinic without tele-rehabilitation.

The tele-rehabilitation group received the treatment with the MediTutor software, the ArmTutor (MediTouch, Netanya, Israel) and 3DTutor biofeedback hardware (MediTouch, Netanya, Israel; Figure 2). The MediTouch system was successfully used in hand rehabilitation [33,34]. The ArmTutor was used for practicing elbow flexion and extension, and the 3DTutor was used for practicing supination and pronation. The software provides visual and auditory feedback while performing different tasks and games. Both devices are strapped to the limb using Velcro straps. The hardware includes wireless electro-optical sensors, which detect joint movement and display them on the screen. The system measures passive and active ROM and movement accuracy in relation to the presented virtual goal and speed. Each session included 15 min of flexion–extension exercises with the ArmTutor device and 15 min of supination-pronation exercises with the 3DTutor device. At the beginning of every game, the system measures active ROM. Each session started with the existing active ROM for 5 min at a low speed, and then the settings of the chosen game were altered during the game by the therapist; thus, the patient had to either increase the ROM in order to reach the target of the game or stay at the same range and increase the speed. The therapist followed the patients' performances. The difficulty level was personally graded to each participant according to personal preferences, pain, and fatigue levels.

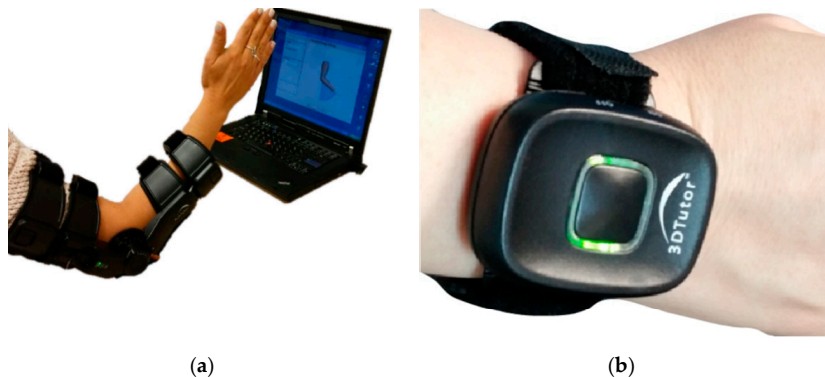

<center>(<b>a</b>)                   (<b>b</b>)</center>

**Figure 2.** The Meditouch hardware used in this study comprise (**a**) the ArmTutor device, used for exercising elbow flexion and extension, and (**b**) the 3-D device, placed and the wrist and used for exercising supination and pronation.

The conventional intervention in the outpatient clinic included exercises of active ROM (flexion, extension, pronation, and supination) of the arm and the elbow. Common active exercises for flexion, extension, and pronation–supination were used. Resistance and range were personally increased for each patient using common intervention accessories, e.g., Theraband and ball. In addition, the conventional therapy sessions included usage of a hot pack, in order to release joint and soft tissue stiffness at the elbow area, and scar massaging.

*2.3. Data Analysis*

Statistical analysis was performed using the Social Sciences Software (SPSS) software (version 21).

Normal distribution was tested using the Shapiro–Wilk test. Since only 5 variables out of 28 were not normally distributed, we present the data as median and interquartile range (IRQ) and performed nonparametric tests to analyze the data.

The percentage of improvement of each measure between the different measurement time points was calculated according to the following equation:

$$Change\ in\ X\ [\%] = \frac{X_{post\ treatment}}{X_{at\ baseline}} \cdot 100 - 100 \tag{1}$$

where $X_{at\ baseline}$ is the measure immediately after the surgery and $X_{post\ treatment}$ is the measure at one of the three post-treatment evaluations.

The Friedman test was used separately in each group to detect differences of the percentage of change in all measures between repeated measurements at different points in time. Where significance was detected, the Wilcoxon test was then used as a post hoc test. The Mann–Whitney U test was used to examine between-group differences. The significance level for all statistical analyses was set at $p < 0.05$. The JASP software (V0.14.1, JASP team, University of Amsterdam, Amsterdam, The Netherlands) was used to perform Bayesian Mann–Whitney analyses, and the BF01 was reported.

**3. Results**

During the first month of treatment, the subjects in the tele-rehabilitation group received a median and IQR of 14.0 (3.5) sessions, of which 8.0 (2.0) were in the clinic and 5.0 (3.5) were via tele-rehabilitation. The self-practice time of the tele-rehabilitation group, recorded by the biofeedback system, was 4.7 (8.9) hours. The subjects in the control group received a median and IQR of 11.0 (3.5) sessions. Overall, for three months following the surgery, the tele-rehabilitation group received 22.6 (5.2) sessions, and the control group received 21.8 (6.2) sessions.

Four subjects (two from each group) had subsequent surgeries (three of them due to limitation in ROM and one due to ulnar nerve entrapment) that occurred after the 3-month evaluation but before the 1-year evaluation. The specific timepoint in which the four subjects underwent additional surgery in relation to the first surgery was 8 and 11 months for the two subjects from the control group and 4.5 and 6 months for the two subjects from the tele-rehabilitation group.

There were no between-group statistically significant differences in measures of passive and active ROM scores for each evaluation timepoint (normalized by the baseline measurement, according to Equation (1); Table S1). The BF01 range was between 1.658 and 2.233, indicating that the results are in favor of the Null hypothesis. However, the number of patients that attained functional ROM results, i.e., active ROM acceptable for Activities of Daily Living (ADLs), following 1 month of intervention was higher in the tele-rehabilitation group. Specifically, functional active flexion ROM (i.e., <130° [35]) was attained by 4 (44.4%) subjects in the tele-rehabilitation group and 1 (11.1%) subject in the control group. Functional active extension ROM (i.e., >30° [35]) was attained by 7 (77.8%) subjects in the tele-rehabilitation group and 4 (44.4%) subjects in the control group. Normal active supination ROM (i.e., >85° [35]) was attained by 7 (77.8%) subjects in the tele-rehabilitation group and 5 (55.6%) subjects in the control group. Normal active pronation ROM (i.e., >75° [35]) was attained by 8 (88.9%) subjects in the tele-rehabilitation group and 5 out of 7 (71.4%) in the control group (data are missing for 2 subjects).

There were no between-group statistically significant differences for each evaluation timepoint (normalized by the baseline measurement according to Equation (1); Table S1) for the grip force (BF01 range: 1.057–2.472), PREE (BF01 range: 0.949–2.415), DASH (BF01 range: 1.314–1.957), and JHFT (BF01 range: 1.921–2.178) scores. However, 8 (88.9%) subjects from the tele-rehabilitation group improved their JHFT score by more than 10.2 points (which is Minimal Clinically Important Difference (MCID) in the JHFT score, found for adults with upper extremity musculoskeletal complaints [36,37]), but only 5 (55.6%) in the control group improved their JHFT score by more than 10.2 points. The PREE and DASH scores for each group in each time point are presented in Figure 3.

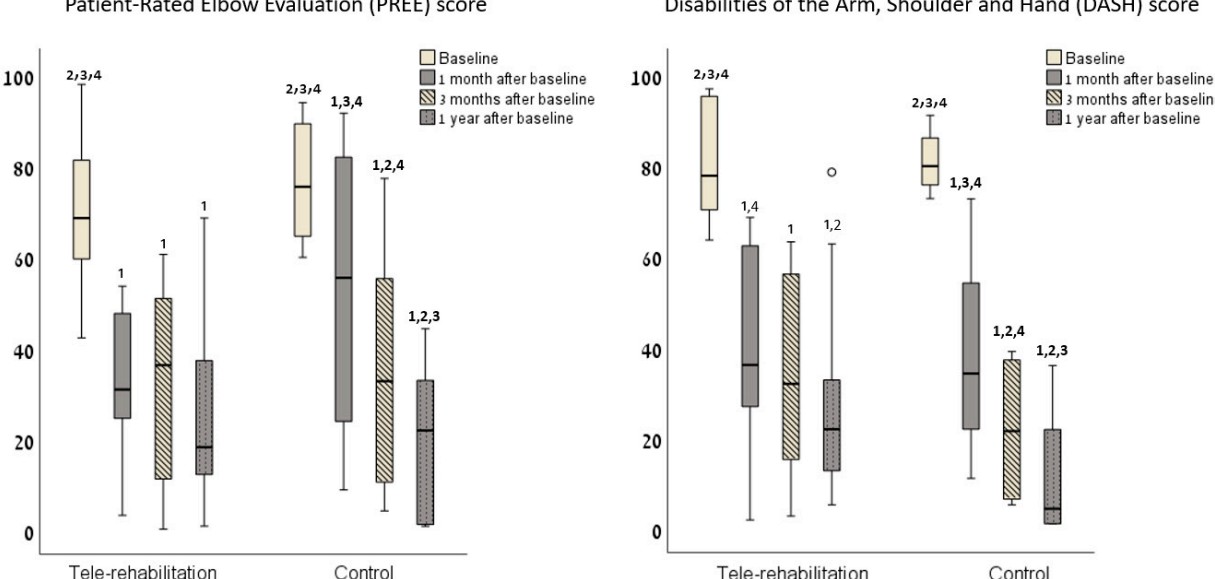

**Figure 3.** The Patient-Rated Elbow Evaluation (PREE) (left frame) and Disabilities of Arm, Shoulder and Hand (DASH) (right frame) scores for each group at baseline (before the intervention) and at the 3 postintervention timepoints. The numbers above each box plot represent statistical significance between timepoints, as follows: [1] Statistically significant differences with the test performed at baseline. [2] Statistically significant differences with the test performed after 1 month. [3] Statistically significant differences with the test performed after 3 months. [4] Statistically significant differences with the test performed after 1 year.

The subjects in the tele-rehabilitation group reported a higher level of enjoyment of the self-practice ($p = 0.046$) and needed less help from a family member during the self-practice ($p = 0.028$). There were no further between-group differences in the answers to the patient satisfaction questionnaire. Additionally, the subjects in the tele-rehabilitation group reported the comfort level of using the biofeedback system as 80.0 (20.0).

## 4. Discussion

We compared tele-rehabilitation treatment outcomes with conventional rehabilitation in individuals post-elbow fractures. While no statistically significant differences were found between the two groups, the number of patients that attained functional ROM following 1 month of intervention was higher in the tele-rehabilitation group compared to the control group. The subjects in the tele-rehabilitation group also reported a higher level of satisfaction and needed less help from a family member during practice.

Sufficient active elbow ROM is important to maintain normal daily activity. In a study that restricted active ROM at the elbow, forearm, wrist, or fingers in healthy subjects, the authors reported that elbow and forearm restriction caused a similar decrease in hand function as restricted active ROM at the wrist and hand [38]. This indicates that the positioning of the limb via the elbow joint is as important to the successful completion of daily tasks as the ROM of the hand fingers. Although there were no between-group differences in the percentage of improvement in the ROM, the number of subjects who attained functional ROM immediately following the intervention was higher in the tele-rehabilitation group compared to the control group, implying that certain tasks of ADLs might have been less demanding for them. This is also supported by the higher number of subjects from the tele-rehabilitation group that improved their JHFT score by more than the MCID in the JHFT score compared to the control group.

Our results suggest that, although not statistically significant, the tele-rehabilitation group showed a tendency toward larger improvement in the PREE and DASH scores immediately following intervention (Figure 3 and Table S1), while in the control group, a slower and constant improvement was apparent over the period of a year. The pain following elbow surgery was found to be a predictor of disability, even when ROM was considerably improved [10]. Pain may occur due to swelling and immobilization, so exercises that incorporate movements usually result in pain relief [4,35]. We therefore assume that the tele-rehabilitation group which reported a high level of usability and a higher level of enjoyment of the self-practice and needed less help from a family member during the self-practice, were able to practice more, thereby alleviating the pain during the intervention.

Overall, this pilot study suggests that both interventions, whether they did or did not incorporate tele-rehabilitation, induced effective outcomes of pain reduction and increased functionality and ROM. There were no dropouts during the intervention, suggesting that the subjects were satisfied with their treatment. Additionally, during the first month of treatment, the subjects in the tele-rehabilitation group received more sessions compared to the conventional rehabilitation group, suggesting that when sessions are available for the patients at the comfort of their home, they choose to schedule these sessions more frequently compared to patients who must arrive at a clinic to receive treatment. It should be noted in this regard that age can be a determinant for the success of tele-rehabilitation, as younger people are more prone to use new technologies. Previous research also reported no differences in the satisfaction levels of individuals who received tele-rehabilitation treatment versus conventional treatment, even when the tele-rehabilitation treatment was not combined with treatment in the clinic, as done in this study [39–42]. Patient satisfaction is an important measure of the efficacy of an intervention plan, affecting the patient's motivation and responsiveness [42,43]. Patient satisfaction is also affected by the therapist–patient relationship [44]. One might expect that this relationship would be impaired when the patient and therapist are situated in different locations and no physical contact exists; however, this was not the case, according to our results.

The first study limitation is the lack of documentation of the self-practice time of the control group. Although they reported that they practiced at home, there was a low response in filling out the self-practice documentation form provided to them. Considering the lack of documentation for self-practice and the higher number of sessions received by the tele-rehabilitation group in the first month of intervention, it might be possible that one group practiced more than the other. This factor might have affected our results. Future studies might therefore strive to compare two groups who received an equal number of sessions and self-practices. This might prove challenging, as the low compliance of individuals with elbow injuries to arrive at the clinic would force the researchers to limit the tele-rehabilitation sessions, thereby restricting the treatment of the tele-rehabilitation group. Second, a large sample size is needed. Last, the PREE and DASH questionnaires relate to some functions that may have been performed in the dominant hand before the injury. Since two thirds of the subjects in each group were injured in their nondominant hand, their scores might have been underestimated. However, as shown by [45], the slight effect of the side of injury on the DASH score is lower than the minimally important difference.

In conclusion, since tele-rehabilitation is a cost-effective treatment suitable for patients with accessibility difficulties or who have difficulty in arriving at the clinic, we conclude that it could be incorporated in the framework of treatment following elbow fractures, as its outcome measures are not different from those acquired during traditional treatment.

**Supplementary Materials:** The following are available online at https://www.mdpi.com/2076-3417/11/4/1708/s1, Table S1: Median and (interquartile values) of the passive and active Range of Motion (ROM) of the elbow, the Jebsen Hand Function Test (JHFT) score, the Patient-Rated Elbow Evaluation (PREE) score, the Disabilities of the Arm, Shoulder and Hand (DASH) score, and grip force, presented as percentage of the baseline values (pre-intervention). P values were calculated separately for each group using the Friedman test. Where statistical significance was found, the Wilcoxin post hoc test was applied.

**Author Contributions:** Conceptualization, N.M.; Data curation, N.M.; Formal analysis, S.P. and Y.L.; Investigation, S.P. and Y.L.; Methodology, N.M., S.P. and Y.L.; Project administration, N.M., R.P. and Y.L.; Resources, R.P.; Supervision, S.P., R.P. and Y.L.; Validation, S.P. and Y.L.; Visualization, S.P.; Writing—original draft, S.P.; Writing—review & editing, N.M., S.P. and Y.L. All authors have read and agreed to the published version of the manuscript.

**Funding:** This research received no external funding.

**Institutional Review Board Statement:** The study was conducted according to the guidelines of the Declaration of Helsinki, and approved by the Institutional Review Board of Tel Aviv University and the Helsinki committee of the Sheba Medical Center (approval number SMC-2296–15).

**Informed Consent Statement:** Informed consent was obtained from all subjects involved in the study.

**Data Availability Statement:** Data shared are in accordance with consent provided by participants on the use of confidential data.

**Conflicts of Interest:** The authors declare no conflict of interest.

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
