# Peer review of "The Efficacy of Tele-Rehabilitation Program for Improving Upper Limb Function among Adults Following Elbow Fractures: A Pilot Study"

_applsci, doi:10.3390/app11041708_

Round 1
Reviewer 1 Report
The topic of the manuscript is very interesting and, as highlighted by the authors, it is very relevant in the current pandemic condition.
Unfortunately I have some major concerns about the manuscript and the design of the study which significantly affect the reliability of the results.
- The introduction is effective in explaining the importance of early intervention and the need of the adoption of telerehabilitation system; on the other hand the analysis of the state of the art it is quite poor. The authors mentioned some examples of previous studies about telerehabilitation but without providing any details. A deeper analysis of literature would improve the quality of the introduction. Moreover, it would be interesting to add some previous researches that used the same telerehabilitation systems of the present work. Is it the first study that involved the MediTutor software, the ArmTutor and the 3DTutor? In case the present study is the first study with such system, the authors might consider to explain the rationale behind the choice to use it.
- Four subjects had subsequent surgeries after the 3 months evaluation and this could have strongly affected the results of the 1-year evaluation. I would suggest either to completely remove the 4 subjects from the study or to remove their scores from the 1-year evaluation to guarantee comparable the results.
- The authors aimed to compare telerehabilitation treatment to conventional treatment but the number of sessions were variable between the two groups. In fact, telerehabiliation group received a median of 14 sessions while the control group received a median of 11 session (line 190-193). A better procedure would have been to fix the same number of sessions for both groups to assure the comparability of the results. Indeed, the better results obtained by the telerehabilitation group could be due to the higher number of sessions performed and not to the different treatment.
- The authors said that the PREE and the DASH relate to the functions performed in the dominant hand but most of the subjects were injured in the non-dominant hand (line 281-283) and thus results can be affected. I would suggest to adopt more strict inclusion criteria considering the injured hand hand-dominance to assure comparable results.
Minor comments:
Line 107-140: I would suggest to list the four questionnaires using a bullet point list to improve the readibility of this paragraph.
Figure 1. The information about the 27 candidates and the 21 participants recruited might not be very relevant for the readers. I would suggest to remove them. On the other hand, it would be also useful to split the block relative to the intervention into two blocks (one for each group) and to add some more information about it (e.g. 1-2 telerehab sessions + 1-2 conventional for the telerehab group and 3-4 conventional session for the other group).
The descriptions of the evaluation scales are very clear and detailed while the intervention is vague, especially the conventional intervention. I think that additional details about the exercises performed are needed, also a figure of the devices used (the ArmTutor and the 3DTutor) would be useful.
Reviewer 2 Report
Comments on “The efficacy of telerehabilitation…“
This paper is clearly written and relevant.
There are some minor defects:
L76-77: The subjects were assigned to one of two groups (Table 1),6 matched according age and type of fracture (olecranon, distal humerus, radial head, or arthrolysis): telerehabilitation group (N=9) and control group (N=9). No statistically significant differences were found in age, sex, and fracture type between the two groups.
If differences between groups has been removed at outset, why should they appear?
L131-132: regarding concerned the assistance needed to 131 use the intervention at home
?
L136-139: The scoring is performed as follows: uneven Items (1,3,5,7,9) are deducted by one point from the subject's selection and even items (2,4,6,8,10) subtract the subject's choice from five. The sum of the items is multiplied by 2.5 and a score is obtained on a scale between 0 and 100.
???
L143 -145: The telerehabilitation group received 1-2 sessions per week of elbow 143 motion at home via telerehabilitation and 1-2 conventional sessions per week in an out-patient clinic
This means some got 4 sessions/week, some 2/week.
L149-150: After one month, both groups continued to receive occupational therapy treatment in the clinic, without telerehabilitation
If so, what is the reason to compare these groups after one year?
L183-184: the normalized measures
??
L186-187: The significance level for all statistical analyzes was set at p<0.05.
The Authors compare two treatments, they want exclude by chance difference between outcomes, while they should ask what is the probability, that outcomes differ (type I and type II errors problem)
L191-192 The self-practice time of the tele-rehabilitation group, recorded by the biofeedback system, was 4.7 (8.9) hours.
What about control group?
L196-198: Four subjects (two from each group) had subsequent surgeries (three of them due to 196 limitation in ROM and one due to ulnar nerve entrapment) that occurred after the 3 197 months evaluation but before the 1-year evaluation.
How this might affect results? How they performed in comparison to others?
L202 -204: There were no between-group statistically significant differences for each evaluation time point (normalized by the baseline measurement according to equation 1; Table S1) for the passive and active ROM scores.
I suggest rewriting.
L204 -205: functional ROM
What is this?
L206-214: Specifically, functional active flexion ROM (i.e., <130° [29]) was attained by 4 (44.4%) subjects in the telerehabilitation group and 1 (11.1%) subject in the control group. Functional active extension ROM (i.e., >30° [29]) was attained by 7 (77.8%) subjects in the telerehabilitation group and 4 (44.4%) subjects in the control group. Normal active supination ROM (i.e., >85° [29]) was attained by 7 (77.8%) subjects in the telerehabilitation group and 5 (55.6%) subjects in the control group. Normal active pronation ROM (i.e., >75° [29]) was attained by 8 (88.9%) subjects in the telerehabilitation group and 5 out of 7 (71.4%) in the control group (data are missing for 2 subjects).
There were no between-group statistically significant differences.
Here is the point: no statistically significant differences means here that these differences might occur by chance, but what is the chance they are true?
L239: functional results
?
L248-258: Although there were no between-group differences in the percentage of improvement in the ROM, the number of subjects that attained ROM functional results immediately following the intervention was higher in the tele-rehabilitation group compared to the control group, implying that certain tasks of ADLs might have been less demanding for them. This is also supported by the higher number of subjects from the tele-rehabilitation group that improved their JHFT score by more than the MCID in the JHFT score compared to the control group.
Our results suggest that, although not statistically significant, the telerehabilitation group showed a trend towards larger improvement in the PREE and DASH scores immediately following the intervention (Fig. 2 and Table S1), while in the control group, a slower and constant improvement was apparent over a period of a year?
Better outcome immediately after intervention may be confounding factor, Authors may check the correlation between early and late outcomes in both groups.
Reviewer 3 Report
The article still needs further developments to be published. Here are my comments and suggestions to improve it:
- I cannot see correctly the tables, I don’t know it is a format problem or why.
- Include at least telerehabilitation and elbow in the keywords.
- Figure 1 could be smaller.
- In the abstract, it says that the tele-rehabilitation group age is 39.8 +- 15.3 and the control group 50.7+-19.3. However, in Table one 33 (27.9) and 60 (37). I guess that this is because the first addresses mean and std and the second median and interquatile diffence. It also says that there is no statistical difference in age, however, from the values it seem that overall the tele-rehabilitation group is younger. In this sense, it would be very interesting, besides clarifying this point, addressing in the discussion section whether the age can be determinant for the success of tele-rehabilitation, as younger people is more prone to use new technologies.
- In the abstract it also says that tele-rehabilitation is given via a biofeedback systems. Please, explicity specify that it is through Medi Touch’s ArmTutor and 3DTutor systems. Besides, in section 2.2.2 a better description of these equipment and photo of the systems would allow the reader to better understand the process of tele-rehabilitation.
- In section 2.2.1, there is brake at line 101 that should not be. Besides the two paragraphs staring with “Three physical…” and “Four questionnaries…” are hard to read. I suggest reorganizing these paragraphs, maybe with bullets or in other way, to make reading more easy
- In figure 2, the label says PREE (right) and DASH (left), but in the figure it is the opposite.
- In table S1. What is the meaning of a,b,c superscripts?
Round 2
Reviewer 1 Report
The authors correctly addressed all my concerns and improved the quality of the manuscript.
Author Response
Thank you for your time and valuable input.
Reviewer 3 Report
The authors have successfully addressed my suggestions.
Author Response

(The authors gave the same response as above.)
